# Quantifying protein densities on cell membranes using super-resolution optical fluctuation imaging

Tomáš Lukeš[1,2], Daniela Glatzová[3,4], Zuzana Kvíčalová[3], Florian Levet [5,6], Aleš Benda[3,7], Sebastian Letschert[8], Markus Sauer[8], Tomáš Brdička[4], Theo Lasser [1] & Marek Cebecauer [3]

Quantitative approaches for characterizing molecular organization of cell membrane molecules under physiological and pathological conditions profit from recently developed super-resolution imaging techniques. Current tools employ statistical algorithms to determine clusters of molecules based on single-molecule localization microscopy (SMLM) data. These approaches are limited by the ability of SMLM techniques to identify and localize molecules in densely populated areas and experimental conditions of sample preparation and image acquisition. We have developed a robust, model-free, quantitative clustering analysis to determine the distribution of membrane molecules that excels in densely labeled areas and is tolerant to various experimental conditions, i.e. multiple-blinking or high blinking rates. The method is based on a TIRF microscope followed by a super-resolution optical fluctuation imaging (SOFI) analysis. The effectiveness and robustness of the method is validated using simulated and experimental data investigating nanoscale distribution of CD4 glycoprotein mutants in the plasma membrane of T cells.

[1] Laboratoire d'Optique Biomédicale École Polytechnique Fédérale de Lausanne, STI-IBI, CH-1015 Lausanne, Switzerland. [2] Department of Radioelectronics, FEE, Czech Technical University in Prague, 166 27 Prague, Czech Republic. [3] Department of Biophysical Chemistry, J. Heyrovsky Institute of Physical Chemistry, Czech Academy of Sciences, 182 23 Prague, Czech Republic. [4] Laboratory of Leukocyte Signalling, Institute of Molecular Genetics, Czech Academy of Sciences, 142 20 Prague, Czech Republic. [5] Interdisciplinary Institute for Neuroscience UMR 5297 CNRS Université de Bordeaux, 33077 Bordeaux, France. [6] Bordeaux Imaging Center, UMS 3420 CNRS Université de Bordeaux US4 INSERM, 33077 Bordeaux, France. [7] Imaging Methods Core Facility BIOCEV, 252 50 Vestec u Prahy, Czech Republic. [8] Department of Biotechnology and Biophysics, Biocenter, University of Wuerzburg, D-97074 Wuerzburg, Germany. Correspondence and requests for materials should be addressed to T.L. (email: theo.lasser@epfl.ch) or to M.C. (email: marek.cebecauer@jh-inst.cas.cz)

Surface molecules influence vital functions of living cells. Proteins form the largest pool among these essential molecules. A growing body of evidence supports the hypothesis that proteins are not distributed homogeneously but rather in complexes, clusters and other higher-order patterns in membranes of cells and organisms[1–4]. It has been experimentally demonstrated that these protein clusters are involved in the regulation of signal transduction and other vital cell processes[5]. It is therefore important to monitor heterogeneous distribution of membrane proteins at nanoscale and with quantitative approaches[6]. The efficiency of currently available tools for cluster analysis is limited in high-density regions by the ability to identify and localize individual molecules in raw images, as well as the blinking properties of the emitters[7, 8]. This has motivated us to develop a robust method for investigation of molecular organization of cell membranes that tolerates diverse experimental conditions.

The size of membrane protein assemblies varies and is frequently smaller than 200 nm, which is below the resolution limit of classical fluorescence microscopy. During the last two decades, super-resolution techniques have been developed that overcome the diffraction limit[9, 10] and provide a detailed view of structures smaller than 200 nm. Single-molecule localization microscopy (SMLM) has been frequently used to characterize membrane protein assemblies[11–14]. SMLM techniques such as (fluorescence) photoactivated localization microscopy (PALM, FPALM)[15, 16] and (direct) stochastic optical reconstruction microscopy (STORM, dSTORM)[17–20] rely on temporal discrimination of otherwise spatially overlapping fluorophore images. In sequences of at least several thousand images, the position of fluorescent molecules is determined by fitting a model function to the imaged point spread functions (PSFs). In high-density regions this fitting procedure may meet its limit, leading to under-counting errors with significant localization errors for overlapping molecules[7]. The stochastic blinking behavior of fluorophores may result in multiple localizations from single molecules[8]. It was previously reported that high photoswitching rates in combination with high emitter densities can give rise to the appearance of artificial clusters[7]. These limitations may compromise the quantification of densely packed proteins in membrane clusters. Characterization of such protein clusters becomes a challenge because current methods for cluster analysis[12–14, 21–25] rely both on difficult-to-

model photophysical properties and on acquisition parameters of the SMLM data. In this work, we readdress these problems with an approach based on SOFI and present an innovative and general method to study molecular distribution on cell membranes that overcomes the aforementioned limitations.

SOFI is an optical super-resolution technique that exploits the spatio-temporal photon traces created by stochastically blinking fluorophores. SOFI disentangles the overlapping PSFs by employing higher-order statistics. The strong temporal cross-correlation over several neighboring pixels is the underlying cause of SOFI super-resolution[26, 27]. The achieved resolution improvement results from the properties of spatio-temporal cross-cumulants calculated from the entire image sequence of 2D[26] or 3D[28] images. SOFI can be used to analyze SMLM data, but tolerates much higher emitter densities[29, 30]. Balanced SOFI (bSOFI) combines the information content of several cumulant orders in a system of linear equations, allowing to extract physical meaningful parameters such as brightness, emitter density and the on-time ratio of the blinking emitters[31]. Molecular density is a calculated parameter based on the full image sequence and not on individual localizations acquired in frame-by-frame data postprocessing. In bSOFI multiple blinking of individual emitters improves the bSOFI signal and, therefore, the accuracy of these statistically estimated parameters.

SOFI used for estimation of molecular parameters shows some similarities in its mathematical formulation with image correlation spectroscopy (ICS) and related methods such as spatiotemporal ICS, raster ICS, and k-Space ICS[32–34]. All these methods exploit temporal correlations over a sequence of images. However, SOFI relies on higher-order cumulants for super-resolution imaging. The molecular parameter maps, resulting from bSOFI, are calculated with increased spatial resolution, but compared to ICS on generally longer time intervals. bSOFI assumes the sample to be stationary during the image acquisition, where the fluctuations in intensity arise mostly from the blinking behavior of observed emitters. In contrast, ICS intends to measure fast molecular processes such as diffusion or number of molecules, but does not provide super-resolution.

Previously, we demonstrated the potential of bSOFI for molecular density estimation. The investigated samples have been tubulin meshworks[31] or dense protein structures such as focal adhesions labeled with fluorescent proteins[30].

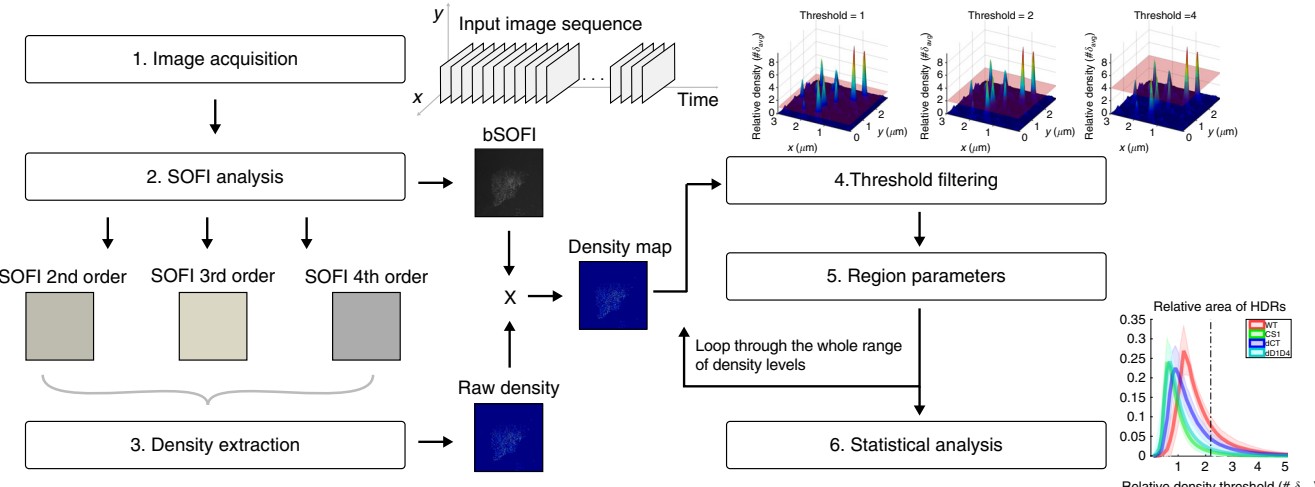

**Fig. 1** The workflow of SOFI-based molecular density analysis. SOFI images of different cumulant orders were calculated and used to extract molecular densities. The background was removed using the bSOFI image as a transparency mask. High-density regions (HDRs) were segmented by varying the threshold parameter over the whole range of available density levels. For each threshold, the area, equivalent diameter, and number of HDRs were extracted and plotted as a function of the density threshold (Fig. 3). The procedure is then repeated for each sample and ROI

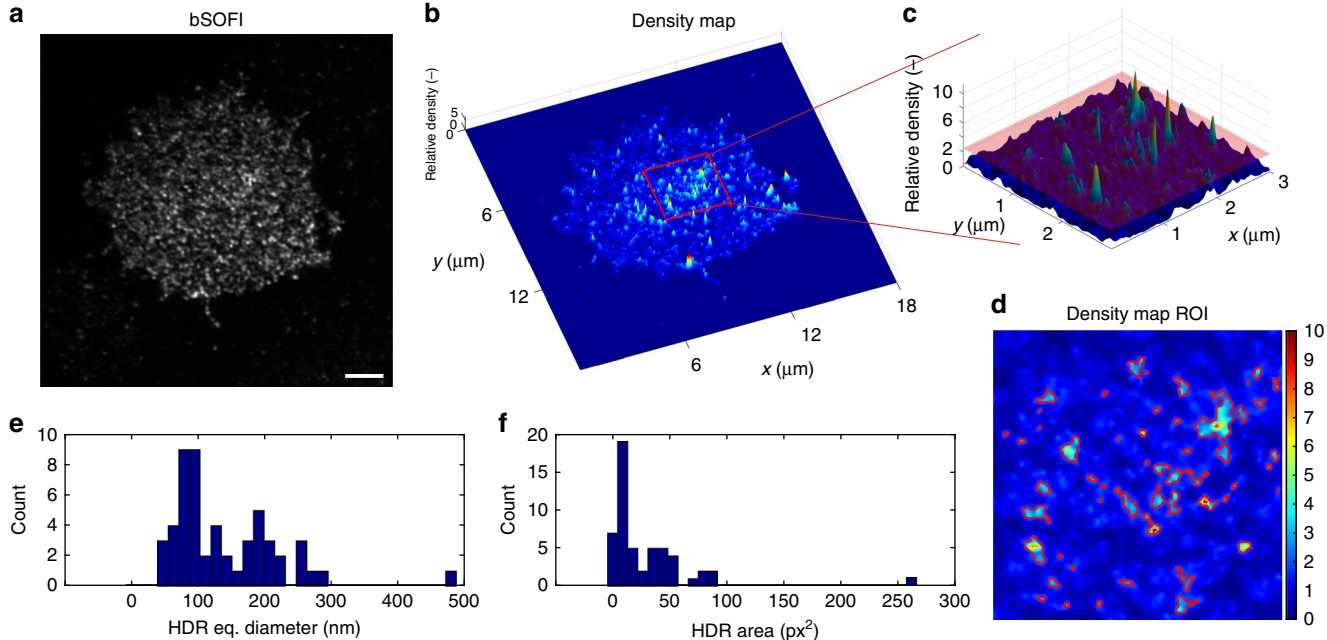

**Fig. 2** Example of data processing for a single cell expressing wild-type CD4-mEos2 fusion protein. First, bSOFI image **a** is generated and a molecular density map **b** is calculated. Segmentation of clusters in the 3 × 3 μm region of interest (ROI) as indicated in **b** is performed for each molecular density by monotonically increasing the threshold (an example is shown for a relative density threshold equal to 2.2 times the mean density (**c**, **d**). For each threshold, a histogram of equivalent diameters (**e**), i.e. diameter of a circle of the same area as the non-circular region, and a histogram of the measured area (px$^2$; **f**) of high-density regions (HDRs) in the ROI shown in **d** are presented

Here we extend this preliminary work substantially for a quantitative and user-independent analysis of protein densities on cell membranes. We present a full framework of a SOFI-based clustering analysis for quantitative assessment of heterogenous protein distributions. By an automatic analysis of the bSOFI density maps, our method quantifies molecular clustering behavior and allows direct comparison of different membrane molecules, their mutant variants or membrane organization at altered experimental conditions (e.g. the effect of inhibitors or drugs).

## Results

**Molecular density clustering analysis.** For quantifying the protein distribution in the plasma membrane of T cells, we acquired image sequences with a total internal reflection fluorescence (TIRF) microscope equipped with an EMCCD camera to detect the fluorescence originating from individual fluorescent emitters (see Methods). The proteins of interest were labelled with adequate blinking fluorophores, i.e. emitters cycling between dark/bright states.

The algorithm work flow is shown in Fig. 1. All acquired image sequences are first drift-corrected with sub-pixel precision. Using ThunderSTORM[35], we measured lateral drift using fluorescent beads (fiducial markers) present in the images. These drift-corrected image sequences were then processed by the bSOFI algorithm using second, third and fourth-order cumulant analysis (see Methods). We extracted molecular density maps by combining the cumulant images in a system of linear equations (see Methods). As shown previously[30], the accuracy of the density calculation is mainly determined by the size of the input image sequence. We acquired image sequences of 5000 frames for each dataset, optimizing the number of frames by analyzing the signal to noise ratio (SNR)[30]. The density maps were further analyzed to extract molecular density and clustering parameters for a direct comparison of tested molecules (see Methods). Figure 2 shows a data-processing example for a single cell.

Instead of setting the threshold by the examiner, our SOFI-based clustering analysis evaluates the density maps by monotonically increasing the threshold in a full spectrum of calculated densities (see Methods). Starting with a low threshold, large regions with a low average density are segmented. Increasing the threshold step by step allows precise density quantification (Supplementary Fig. 1). The algorithm analyzes each region of interest (ROI) by calculating, for each density threshold, the average number and area of HDRs, as well as the relative area occupied by the HDRs. The averaged data across all cells for each protein variant over the range of density thresholds are shown in Fig. 3a–c. This analysis provides an overview of HDR parameters in relation to the density threshold, unraveling the overall clustering behavior of the samples under study. Inset images in Fig. 3a and Supplementary Fig. 1 indicate how the density threshold affects the detection of HDRs. Detailed statistics of the quantitative molecular density data can be further presented for the optimal density threshold (Fig. 3d) or any other threshold selected, for example, based on biological reasoning. The optimal density threshold shown in Fig. 3a–c is determined automatically by the algorithm (Supplementary Fig. 2) to distinguish between clustering that corresponds to a random distribution of proteins and the non-random behavior of proteins that creates larger HDRs. This fully automated algorithm does not require manual selection of this important parameter by the user. When calculating the fourth-order SOFI image, the pixel size of the resulting SOFI density map is 26.25 nm. According to the Shannon-Nyquist sampling theorem, the smallest detectable HDR would have a diameter of 52.5 nm. The density analysis can distinguish differences in HDR diameters in increments of 26.25 nm. Higher resolution could be possible with higher-order SOFI images at the expense of more input images, i.e. longer acquisition times. To verify our approach in membrane areas with different molecular densities, we performed simulations that indicate good reliability of the analysis across a broad range of HDR densities (500–3000 mol/μm$^2$) and HDR to background

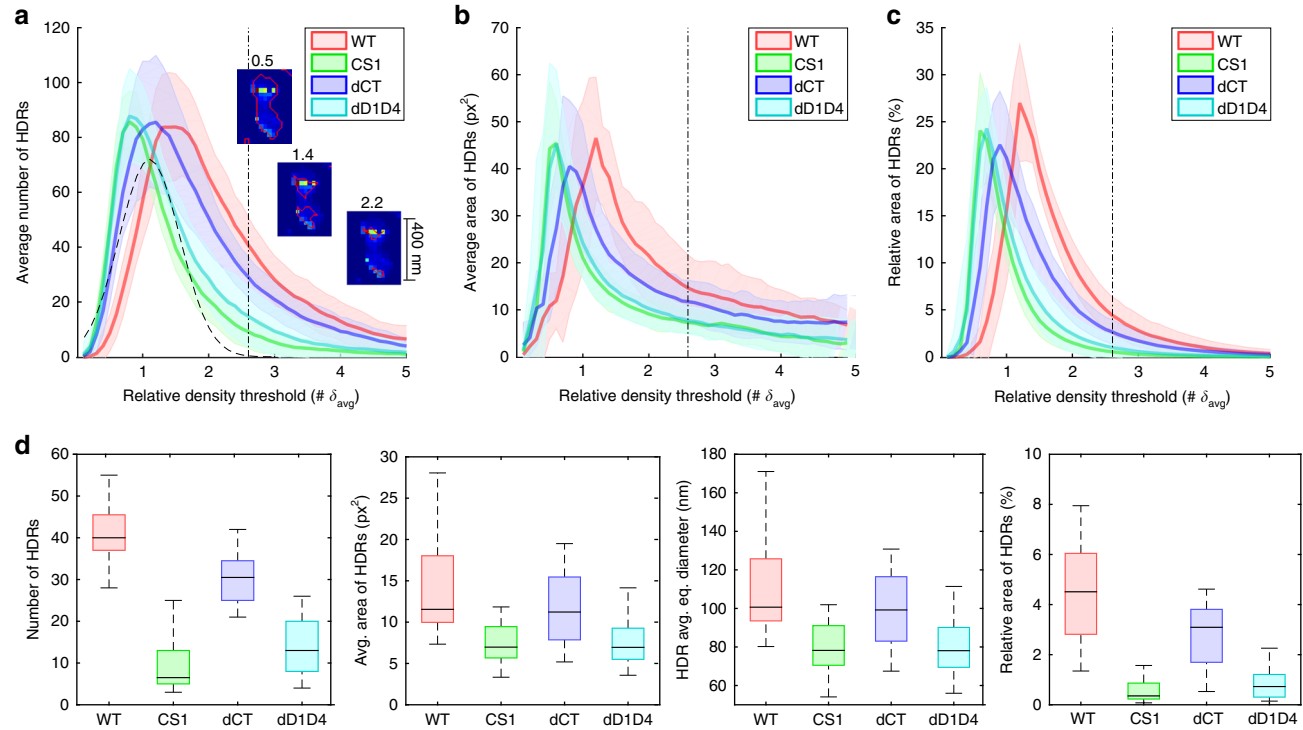

**Fig. 3** SOFI analysis of four CD4 protein variants in resting T cells immobilized on poly-L-lysine-coated coverslips. Native CD4 (WT), palmitoylation mutant (CS1) and variants lacking the extracellular (dD1D4) and cytosolic parts (dCT) were tested ($n = 20$ per variant). **a** Number of high-density regions (HDRs) averaged over all samples for each CD4 variant. Density thresholds are related to the mean density calculated over the $3 \times 3\,\mu m$ ROIs of all samples. The inset images show examples of the segmented HDRs for various thresholds indicated above the image. **b** HDR area averaged over all samples for each CD4 variant in $px^2$, where pixel size is 25 nm. **c** Relative area occupied by HDRs related to the total area of the ROI. **d** Box plots of the properties of HDRs for a threshold equal to 2.6 $\delta_{avg}$ (marked by the vertical dash dot line). The chosen threshold is the value, where Gaussian function of a random distribution (marked in **a** by the dashed line) falls below 1 (Supplementary Fig. 2). Semi-transparent color areas in **a**–**c** represent standard deviation. In each box plot in **d**, the box represents the interquartile range (IQR), the central mark is the median, and the whiskers extend to the most extreme data points. Each box plot was calculated over 20 samples

ratios (20–100). The accuracy of HDR detection increases with increasing HDR to background ratio, which was sufficiently high for all conditions in our real cell experimental data (see Methods and Supplementary Fig. 3).

**Protein nanoscale organization**. In order to validate our approach, we expressed four different mutant variants of mEOS2-labeled CD4 and analyzed individual protein distributions on the plasma membrane of resting T cells immobilized on poly-L-lysine-coated glass coverslips. Using TIRF microscopy, we imaged 20 cells for each CD4 variant (i.e. 80 in total), acquiring 5000 frames per cell. Tested mutants were native CD4 protein (WT), palmitoylation mutant (CS1), and truncated variants lacking the extracellular (dD1D4) and cytoplasmic (dCT) domains (Supplementary Fig. 4).

Segmentation of SMLM data acquired for CD4(WT)-mEos2 indicated the accumulation of native CD4 in HDRs with irregular shape, frequently forming networks (Fig. 2 and Supplementary Fig. 5). SMLM-based cluster analysis of these localization data would be a challenge due to the limitations discussed above and in Supplementary Fig. 5. Moreover, we compared our SOFI-based approach with the state-of-the-art clustering analysis based on SMLM data and showed that SOFI exhibited a consistent performance across a whole range of tested conditions with no cluster-like artifacts (Supplementary Figs 6 and 7). On the contrary, SMLM-based clustering analysis is prone to artifacts if the optimal conditions are not met (i.e. in the case of low irradiation intensity, short off times of fluorophores, high molecular density per frame). SOFI-based clustering analysis

achieved an unbiased robust performance and comparable results under different experimental conditions, which underlines the "quantitativeness" of our approach (Supplementary Figs 6 and 7).

Thus, we used our SOFI-based clustering analysis to quantify distribution of CD4 variants on the surface of resting T cells. To minimize cell-size dependency and aiming for a true comparative protein density analysis among different CD4 variants, we selected a $3 \times 3\,\mu m$ ROI in each cell. All CD4 variants exhibited comparable protein expression levels (Supplementary Fig. 8). Figure 3 summarizes the quantitative data on CD4 membrane organization and indicates significant differences between the tested protein variants at the cell surface of resting T cells. As shown in Fig. 3d, native (WT) CD4 are organized in HDRs covering a large part of the plasma membrane as indicated in Roh et al.[36]. Such arrangements depend on the intact extracellular domain and palmitoylation of CD4 since mutants lacking these structures exhibit more random distribution with rare accumulation in rather small HDRs (Fig. 3d). This observation is also supported by the truncation variant, which lacks the cytoplasmic part (dCT) but contains a single cysteine residue for palmitoylation. Reversibility of this protein lipid modification can lead to a more rapid depalmitoylation of dCT variant compared to native CD4 with two membrane-proximal cysteine residues (Supplementary Fig. 4). Indeed, dCT variant forms fewer HDRs than native CD4, but is more heterogeneously distributed compared to a non-palmitoylatable mutant (CS1).

The results presented in Fig. 3 strongly indicate the ability of our new method to identify HDRs with irregular shape and varying densities. The imaged cells also exhibited a high level of

intercellular variability, especially in case of the intermediate phenotype (dCT), and heterogeneity between HDRs identified within ROIs (Supplementary Fig. 9). The presented results of clustering analysis performed under diverse (real) conditions emphasize the robustness of our method.

## Discussion

In this work, we introduced a novel method for the characterization of molecular organization and clustering on cellular surfaces. The method shows an unbiased performance over a broad spectrum of non-ideal experimental conditions, which are common in single-molecule localization microscopy and microscopy in general. Our fully-automated quantitative clustering analysis is based on SOFI, which provides several distinct advantages over SMLM-based approaches: (i) applicability to densely populated regions (overlapping fluorescence emitters) without a need to adapt specific imaging conditions[37], which are not always accessible; (ii) no need for multiple blinking corrections[38, 39] or other specialized approaches for circumventing the problem of cluster artifacts generated by overcounting of blinking fluorophores[11, 40], and (iii) inherent access to molecular density without a priori assumptions about the clustered molecules. Our approach does not require molecular localization coordinates to calculate clustering properties of proteins (or other molecules) on the cell surface. Importantly, the algorithm provides quantitative molecular density analysis of protein distributions independent of any user-defined parameters. The optimal density threshold to distinguish a random distribution of proteins from clustering is automatically determined for a tested dataset. We demonstrated the applicability of the proposed method by analyzing the surface distribution of CD4 glycoprotein, which forms large, dense, and interconnected regions on human T cells. Our molecular density analysis indicates the importance of the extracellular domain and of receptor palmitoylation for the organization of CD4 on the plasma membrane. This key conclusion is based on the analysis quantifying the clustering behavior of proteins across all density levels available in the sample with no need of an arbitrary threshold specified by the user (Fig. 3a). The fully automated approach opens widely the door for the identification of molecules, variants, or experimental conditions where diverse supramolecular structures are formed at different densities. These would be more difficult in a cluster analysis depending on the manual threshold selection.

We believe that the presented method represents an innovative and universal tool to study molecular distribution on cell membranes. It bears the potential to be extended to any surface molecules accessible for fluorescent labeling under physiological, pathological, or pharmacological conditions.

## Methods

**Microscope setup**. A customized setup built on an inverted optical microscope (IX71, Olympus) was used for cell imaging. A 150 mW 561 nm laser (Sapphire, Coherent) and a 100 mW 405 nm laser (Cube, Coherent) provided the excitation and activation, respectively. An acousto-optic tunable filter (AOTFnC-400.650-TN, AA Optoelectronics) provided fast switching of the laser sources. Both lasers were combined and focused into the back focal plane of an objective (UApoN 100x, NA = 1.49, Olympus). Total internal reflection was achieved with a commercial TIRF module (IX2-RFAEVA-2, Olympus) and the fluorescence emission was detected by an EMCCD camera (iXon DU-897, Andor).

**Generation of CD4 mutant variants**. pXJ41-mEOS2 plasmid was prepared by cloning the full mEOS2 sequence[41] (kind gift of Sean A. McKinney; Janelia Farm, Ashburn, VA) to the pXJ41 vector. The pXJ41-mEOS2 plasmid includes 5′UTR and the leader sequence of human CD148 with c-Myc tag for better protein expression in Jurkat T cells and immunolabeling.

The coding sequence of native CD4 and its non-palmitoylatable variant (CS1; mutations C419S and C421S) was a kind gift of W. Popik (Meharry Medical College, Nashville, TN;[42]). cDNA was amplified using primers TATGGTACCAAG

AAAGTGGTGCTGGGCAAAAAA and GGATCCAATGGGGCTACATGTCTTC TGAAACC and sub-cloned into pXJ41-mEOS2 vector. Primers TATGGTACCA AGAAAGTGGTGCTGGGCAAAAAA and GGATCCACAGAAGAAGATGCC TAGCCCAAT were used to generate CD4$_{1-419}$ variant lacking the intracellular part (dCT) and GATCCGGAGGTGGATCTAGTGCGATTAAGCCAGACATGAAG and CTCGAGTTATCGTCTGGCATTGTCAGGCAATC for CD4$_{387-458}$ variant lacking the extracellular part (dD1D4). The products were subcloned into pXJ41-mEOS2 using *Kpn*I and *Bam*HI restriction sites.

**Sample preparation**. Jurkat T cells in RPMI-1640 media (Sigma-Aldrich), complemented with glutamine and 10% fetal calf serum (Life Technologies), were grown in an incubator under controlled conditions of 37°C, 5% $CO_2$, and 100% humidity. The cells were transiently transfected using the Neon® transfection system (Life Technologies). One microgram of vector DNA per shot (3 pulses of 1325 V lasting for 10 ms) per 200,000 cells was used (see manufacturer's instructions). Twenty-five-millimeter diameter microscope coverslips were cleaned by incubation with 2% Hellmanex (Sigma-Aldrich) overnight at 42°C and subsequently washed with MiliQ water. Prior to use, the coverslips were coated with poly-L-lysine (Sigma-Aldrich). Twenty hours after transfection, the cells were washed with PBS, resuspended in phenol red-free RPMI-1640 media (Sigma-Aldrich), seeded on the poly-L-lysine-coated coverslips, and incubated for 5 min at 37°C under 5% $CO_2$. After a quick PBS wash the cells were fixed using 4% paraformaldehyde in PBS at 37°C for 10 min. After removal of excess liquid, the fixation was stopped with 0.1 M $NH_4Cl$ in PBS and the cells were washed with PBS. Finally, the coverslip was placed into a ChamLide holder for imaging (Live Cell Instruments).

**Imaging**. Fixed cells were imaged in a PBS solution at room temperature. For monitoring drift, 200 nm gold beads (BBI international) were added to the sample. The mEos2 fluorophore was excited at 561 nm with a maximum power of ∼ 30 mW and activated by a 405 nm laser with a maximum power of ∼ 0.3 mW (both measured at the sample plane). Cells were imaged with an EMCCD camera using an EM gain of 300 and an exposure time of 32 ms.

**SOFI molecular density analysis**. SOFI needs the sample to be immobile during image acquisition, and imaging beyond the diffraction limit demands drift correction. Tracking the positions of gold nanoparticles provides translational motion vectors in between consecutive frames. Registering consecutive frames with subpixel precision using bilinear interpolation was used for drift correction.

The drift-corrected image sequence was sub-divided into sub-sequences of 500 frames each. These sub-sequences were chosen sufficiently short in order to minimize the influence of photobleaching[22, 24]. In each subsequence, SOFI images of second, third and fourth-order were calculated. These SOFI images were then averaged across all sub-sequences.

The SOFI-based molecular density analysis was programmed in MATLAB taking into account a linearization procedure as described in ref.[24]. Combining SOFI images of different orders allows one to extract density maps (Supplementary Note 1).

Molecular density (i.e. number of emitters per pixel area) at pixel position **r** is given as

$$N(\mathbf{r}) = \frac{g_2(\mathbf{r})}{(3K_1^2 - 2K_2)^{\frac{3K_1^2 - K_1\sqrt{3K_1^2 - 2K_2} - 2K_2}{2(3K_1^2 - 2K_2)}}\left(1 - \frac{3K_1^2 - K_1\sqrt{3K_1^2 - 2K_2} - 2K_2}{2(3K_1^2 - 2K_2)}\right)}, \text{where } K_1(\mathbf{r}) = \frac{\mu_2 g_3}{\mu_3 g_2}(\mathbf{r}),$$

$K_2(\mathbf{r}) = \frac{\mu_2 g_4}{\mu_4 g_2}(\mathbf{r})$, $g_2$, $g_3$, $g_4$ represents cumulant images of second, third and fourth-order, respectively. $\mu_n = \mathcal{E}_V\{U^n(\mathbf{r})\}$, where $\mathcal{E}_V\{U^n(\mathbf{r})\}$ is the expectation value of the PSF ($U^n(\mathbf{r})$) of the $n^{\text{th}}$-order cumulant image. For more details, see Supplementary Note 1.

Areas containing only background are removed using the bSOFI image as a mask. The threshold filtering procedure is described in more detail in Supplementary Fig. 1. The algorithm loops through a whole range of density levels presented in the sample, starting with a threshold equal to zero and increasing the threshold step by step in each iteration. For each threshold, the data are further processed to acquire quantitative parameters describing regions with local protein density above the threshold, i.e. high-density regions (HDRs). The algorithm calculates the area, equivalent diameter, number of HDRs, and the proportion of the total area of the ROI covered with HDRs. As a result, we obtain a graph that describes the dependence of HDR parameters on the molecular density and reveals the overall clustering behavior of the sample under study. The reliability of the algorithm was investigated under a broad range of simulations (Supplementary Fig. 3).

The absolute values of SOFI molecular density map depend on expression of the fluorescent markers and parameters of the microscope (particularly excitation intensity and a camera gain). Therefore, we use relative densities and investigate relative changes of local density. For simulations, the molecular density maps were normalized by the mean density calculated over all ROIs. For the experimental data, samples were split into four groups according to four CD4 variants. In each group, the group mean density was calculated across ROIs of 20 samples. ROI of each sample was first normalized to the same mean within the group and then by the maximum of all group means. This normalization procedure largely removes

the expression dependence in between the experiments while preserving the differences in relative density in between the CD4 variants. Therefore, normalized SOFI density values can be compared across experiments, as long as the parameters of image acquisition remain the same.

**Simulations**. The simulation assumed photokinetics as known for fluorescent proteins in PALM experiments[24]. A photon time trace for each fluorophore was simulated providing the number of emitted photons over time. The pixel intensity at a given time point corresponds to the integration over the brightness originating from fluorophores in the conjugated object localizations. The number of converted photo-electrons was estimated by a Poisson distributed random distribution. The average value was taken as a pixel value multiplied by the detection efficiency. Additive noise corresponding to thermal noise, read-out noise, and gain variations was added as a Gaussian noise contribution. Optical system and camera parameters are matched to the microscope system settings (NA, wavelength, magnification, pixel size etc.).

The ground truth object is composed of 10 HDRs randomly distributed over an area of $3 \times 3 \, \mu m$. The diameter of generated HDRs varies over the range 60–180 nm, whereas the molecular density in HDRs varies over the range 500–3000 molecules/$\mu m^2$. In between the HDRs, individual molecules were randomly distributed such that the HDR to background ratio was {20, 50, 100}. For each test scenario, we simulated a random distribution of labeled molecules, including no clusters as a control. In total, we generated and analyzed 720 simulated image sequences (Supplementary Fig. 3). The simulation proves that the algorithm performs well under a broad range of conditions.

**Data availability**. All data and code are available from the corresponding authors upon request.

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

## Acknowledgements

We would like to thank Guy Hagen for his helpful remarks and comments. We acknowledge Peter Kapusta for technical assistance and professional advice. This work was funded by Czech Science Foundation (M.C.: 15–06989S; https://gacr.cz/en/). T.La. acknowledges the partly funding of the Horizon 2020 project AD-gut (SEFRI 16.0047, H2020-NMP-2015-two-stage − GA 686271), FP7-HEALTH-2013-INNOVA-TION-1 (HEALTH-E2-2014-602812) and the Swiss National Science Foundation (SNSF,

http://www.snf.ch/) under grants 200020-159945 and 205321-138305. T.Lu. acknowledges a SCIEX scholarship (13.183) and a CTU student grant (SGS16/167/OHK3/2T/13).

## Author contributions

T.Lu., T.La. and M.C. conceived the study. T.Lu. and T.La. developed the molecular density analysis. D.G. and Z.K. prepared the samples. D.G. and T.Lu. performed the experiments, T.Lu., F.L., Z.K. and S.L. analyzed the data. A.B. made the microscope setup. A.B., T.B. and M.S. provided research advice. T.Lu., T.La. and M.C. wrote the paper. All authors reviewed and approved the manuscript.

## Additional information

**Competing interests:** The authors declare no competing financial interests.

