## [Peer Review File · Nature Communications]

Reviewers' comments:

Reviewer #1 (Remarks to the Author):

In this study, the authors describe a super-resolution optical fluctuation imaging (SOFI) based strategy to characterize the nanoscale (sub-diffraction) spatial organization of proteins on mammalian cell membrane. The authors present a framework that uses density thresholds to identify regions of high density of proteins (HDRs) in the SOFI-derived density map, and evaluate the size (area and radius) and number of protein-HDRs present in wild-type and mutant CD4 expressing cells. Even though the overall strategy presented here can be useful for interrogating protein organization on plasma membrane, it should be noted that the theoretical concepts behind SOFI, different variations of SOFI-based experimental strategies, as well as the analysis methodologies and mathematical derivations used in this study have already been published and discussed in detail elsewhere in previous papers. In fact, the quantitative method described by the authors is not at all novel, with the mathematical calculations and the concepts being described in a previous publication. The authors basically applied the previously published methodology to look at CD4 organization, however the biological experiments are very preliminary, providing practically no significant biological insight. Additionally, various clustering analysis algorithms of single molecule data have been developed and published in the past few years, these methods can be used to identify nanoscale regions of high protein-density regions and obtain cluster parameters evaluated in this study. Although it is true that precise and accurate identification of actual number of protein molecules in highly dense samples pose certain problems during quantitative analysis of single molecule localization –based methods, but the relative density of molecules (since molecular localizations can be assumed to be proportional to actual number of molecules) can be ascertained pretty accurately with the available analysis techniques, so regions of differential densities can be identified and visualized by single molecule based techniques as well. It is not clear that the SOFI-based strategy discussed here provides any improvement over single-molecule based methods in terms of the parameters evaluated in this study. Overall the publication lacks novelty and does not provide either any analysis or experimental advancement nor does it provide any biological functional insight of any significance. I would not recommend acceptance of the study in Nature Communications.

Reviewer #2 (Remarks to the Author):

The manuscript deals with a very important problem in fluorescence microscopy and super-resolution microscopy of biological cells: The quantification of molecular concentrations in cell membranes. Despite the huge progress that has been made by single-molecule localization microscopy (SMLM) such as PALM or STORM, which allows for identifying and localizing single molecules in biological samples, correct counting of the number of molecules per area remains highly challenging due to the huge variation in molecular brightness and the totally stochastic character of photo-switching, which lies at the basis of SMLM. Here, the manuscript of Lukeš et al. presents a very interesting and promising alternative which is based on Super-resolution Optical Fluctuation Imaging or SOFI. In contrast to SMLM, SOFI measures the fluctuation amplitude via the computation of higher order temporal cumulants of a recorded movie of images, which in principles contains information about intensity fluctuation dynamics of the used fluorescent labels, their local density, and average brightness. Thus, SOFI is completely independent on localizing individual molecules, and works well even at high local label densities. Thus, the paper deals with an important topic and offers a very original and potentially powerful solution to the long-standing problem of quantitative microscopy. There are a few questions/comments which should be answered/commented prior to publication:

1. The authors apply a rather cumbersome threshold approach to estimate size and number of molecules of high density regions. Why not simply plot the molecular density as given by the equation for $N(r)$ on page 4? Does SOFI not directly provide the higher resolution for such a map?
2. The Gaussian fits for the data in supplementary figure 2 do not look very convincing, in particular the double Gaussian fit in panel C. Why not try something like a multiple Poissonian distribution? Also, why are the first 4-5 data points in panel C at the edge of the std-interval?? Should they not be in the middle of the std-interval?
3. As far as I understand, the whole analysis and in particular the simulations are based on the assumption that all molecules follow a simple on-off Bernoulli statistics in their blinking with fixed values of on- and off-switching rates. In reality, one could expect that different fluorescent molecules follow different blinking statistics, or show even multi-step blinking? How would that alter the analysis and its results? From the bSOFI analysis, one gets the on- and off-rates as a spatially dependent function. It would be interesting to show the images for $\rho_{\text{on}}(\mathbf{r})$ and $\rho_{\text{off}}(\mathbf{r})$.
4. The presented work of using cumulants for obtaining number and brightness images has some relation to the work number-and-brightness analysis using image correlation spectroscopy (ICS) introduced by Wiseman, Gratton and Digman. It would be fair to discuss the similarities and

differences that the number-and-brightness analysis by ICS has with the bSOFI approach.

Reviewer #3 (Remarks to the Author):

This is a brief article where the authors employed super-resolution optical fluctuation imaging (SOFI) method to analyze the clustering behaviors of membrane proteins. They examined the expression of wild type and three mutants of CD4 molecules expressed on resting T cell membranes to demonstrate the versatility of the described technology.

Here is a list of things recommended to be revised before publication:

1. It is unclear how much the reported technique is novel and whether it generates significant contributions to the field, as ref. 22, 24, and 25 cited in the manuscript already have demonstrated that SOFI can be applied for imaging and analysis of proteins and their structures on live cells. The introduction part of the manuscript should be revised to better highlight their unique contributions, compared to these prior works.
2. Indeed single molecule localization microscopy (SMLM) has its own limitations in cluster analysis, which is well demonstrated in Supplementary Figure 5. However, there are some previous articles where more rigorous and prudent approaches had been developed to guarantee the integrity of cluster analysis using SMLM data, for example, Rossy, J. et al. Conformational states of the kinase Lck regulate clustering in early T cell signaling *Nature Immunology* 14, 82–89 (2013). I suggest the manuscript should be revised to compare the pros and cons of SMLM and SOFI based methods in more detail. For example, can SOFI based cluster analysis (proposed method in this manuscript) deduce the absolute number of molecules within each cluster?
3. Did the authors verify that the expression levels of wild type and other mutant variants were equivalent to each other? In order to compare the clustering behaviors of these variants (Fig 3, Supplementary Figure 6), the expression level of these variants should be comparable to each other. One of the rudimentary way to show the comparable expression level would be showing a collection of TIRF images acquired for the same acquisition period (up to a few seconds) for each group (or combined raw images without applying density threshold).
4. According to the data presented in Fig 3, the palmitoylation mutant (CS1) molecules are less clustered than the truncated mutant molecules that lack the whole cytoplasmic domain (dCT). More discussion is warranted.

Response to the reviewers' comments

We would like to thank the reviewers for their constructive comments which have been a motivation to revise the manuscript and to initiate additional experiments which strengthened the novelty and robustness of our initial work.

We would like to emphasize the two main changes of our revised manuscript:

- 1) We carefully rewrote the manuscript to increase the novelty of our method. Previously, we demonstrated the potential of bSOFI for molecular density estimation. In this manuscript, we present a full framework of a novel SOFI-based clustering analysis for quantitative assessment of heterogeneous protein distributions. By an automatic analysis of the bSOFI density maps, our method quantifies clustering behavior of the sample and allows direct comparison of clustering of different membrane molecules, their mutants or molecule(s) under diverse experimental condition.

We modified the Introduction to point on the specific differences of our novel method in regard to previous work. In fact, we offer now a new, automatized, SOFI-based clustering analysis.

- 2) We performed a thorough comparison of our SOFI-based approach with the state-of-the-art clustering analysis based on SMLM over a broad range of conditions. A recognized dSTORM dataset (Burgert et al. Histochem Cell Biol, 2015) showing the facts and artifacts of the classical SMLM-based clustering analysis was used for this comparison to avoid bias results. This dataset has been made available by Prof. M. Sauer and his team (Univ. Wurzburg, Germany; now included as additional co-authors). All these new results have been added to the Supplementary Information as Supplementary Figures 6 and 7. At optimal experimental conditions (illumination intensity, molecular density etc.), both methods generated comparable results. However, and in contrast to SMLM, SOFI showed a consistent performance across a whole range of tested conditions avoiding cluster-like artifacts. These data and their analysis support the robustness of our SOFI-based clustering analysis and manifest its potential to be an artifact-free super-resolution method-of-choice for an unbiased quantitative assessment of biological samples. The manuscript was revised to highlight these novel results and features.

Three new figures have been added to the Supplementary Information and the text of the manuscript was revised to strengthen or clarify points raised by the reviewers. In addition, we added a new section to Methods describing the construction of DNA plasmids encoding CD4 variants.

We believe that these new results and the revised text underline the advantages of our novel SOFI-based clustering analysis. Detailed answers to all reviewers' comments follow. All changes introduced to the manuscript have been highlighted in yellow.

Reviewer 1:

In this study, the authors describe a super-resolution optical fluctuation imaging (SOFI) based strategy to characterize the nanoscale (sub-diffraction) spatial organization of proteins on mammalian cell membrane. The authors present a framework that uses density thresholds to identify regions of high density of proteins (HDRs) in the SOFI-derived density map, and evaluate the size (area and radius) and number of protein-HDRs present in wild-type and mutant CD4 expressing cells. Even though the overall strategy presented here can be useful for interrogating protein organization on plasma membrane, it should be noted that the theoretical concepts behind SOFI, different variations of SOFI-based experimental strategies, as well as the analysis methodologies and mathematical derivations used in this study have already been published and discussed in detail elsewhere in previous papers. In fact, the quantitative method described by the authors is not at all novel, with the mathematical calculations and the concepts being described in a previous publication. The authors basically applied the previously published methodology to look at CD4 organization, however the biological experiments are very preliminary, providing practically no significant biological insight.

We are very thankful for finding our approach interesting. We agree that biology-oriented work would require functional assays to show the importance of the changes for the function of CD4 co-receptor. Nevertheless, these application-oriented studies rely on the quality of experimental and computational methods for data acquisition and analysis. This motivated us to develop a new, robust, SOFI-based clustering algorithm and offer it to a broad audience of Nature Communications.

We understand that the novelty of our method has not been expressed properly in the original manuscript. The literature on applications of SOFI (especially bSOFI) is rather scarce (e.g. compared to SMLM). Since no specific article is mentioned in this comment, we can only speculate that the reviewer has in mind references 22, 24 and 25 (in the original manuscript). All these articles mention accessibility of parameters such as molecular brightness in bSOFI calculations. On the other hand, these were not explored to provide quantitative information on membrane proteins. If in the original bSOFI article, S. Geissbuehler and colleagues (S. Geissbuehler et al. *Biomed Opt Express* 2012; ref. 25) focus on the improvement of imaging resolution and balance of image contrast by linearization of the brightness and blinking response, H. Deschout and colleagues (H. Deschout et al. *Nat Comm* 2016; ref. 24) combine PALM and SOFI to improve spatial and temporal resolution for live-cell imaging of focal adhesions. Finally, S. Geissbuehler et al. *Nat Comm* 2014 (ref. 22) focuses on three-dimensional imaging of tubulin networks. None of these articles provides a quantitative method (or full, automatic algorithm) for clustering analysis of membrane proteins. The text of the manuscript was modified to underline and clarify the novelty of our approach.

Additionally, various clustering analysis algorithms of single molecule data have been developed and published in the past few years, these methods can be used to identify nanoscale regions of high protein-density regions and obtain cluster parameters evaluated in this study. Although it is true that precise and accurate identification of actual number of protein molecules in highly dense samples pose certain problems during quantitative analysis of single molecule localization –based methods, but the relative density of molecules (since molecular localizations can be assumed to be proportional to actual number of molecules) can be ascertained pretty accurately with the available analysis techniques, so regions of differential densities can be identified and visualized by single molecule based techniques as well. It is

not clear that the SOFI-based strategy discussed here provides any improvement over single-molecule based methods in terms of the parameters evaluated in this study.

We provide evidence that SMLM-based approaches are less suitable and robust for the analysis of highly dense structures under diverse experimental conditions (Supplementary Figs. 5, 6 and 7). Please, see also the response to the Comment 2 of the Reviewer 3 for more details.

Reviewer 2:

The manuscript deals with a very important problem in fluorescence microscopy and super-resolution microscopy of biological cells: The quantification of molecular concentrations in cell membranes. Despite the huge progress that has been made by single-molecule localization microscopy (SMLM) such as PALM or STORM, which allows for identifying and localizing single molecules in biological samples, correct counting of the number of molecules per area remains highly challenging due to the huge variation in molecular brightness and the totally stochastic character of photo-switching, which lies at the basis of SMLM. Here, the manuscript of Lukeš et al. presents a very interesting and promising alternative which is based on Super-resolution Optical Fluctuation Imaging or SOFI. In contrast to SMLM, SOFI measures the fluctuation amplitude via the computation of higher order temporal cumulants of a recorded movie of images, which in principles contains information about intensity fluctuation dynamics of the used fluorescent labels, their local density, and average brightness. Thus, SOFI is completely independent on localizing individual molecules, and works well even at high local label densities. Thus, the paper deals with an important topic and offers a very original and potentially powerful solution to the long-standing problem of quantitative microscopy.

We greatly appreciate this positive evaluation of our work.

There are a few questions/comments which should be answered/commented prior to publication:

1. The authors apply a rather cumbersome threshold approach to estimate size and number of molecules of high density regions. Why not simply plot the molecular density as given by the equation for $N(r)$ on page 4? Does SOFI not directly provide the higher resolution for such a map?

Yes, SOFI provides high resolution density maps. In order to describe clustering behavior of tested proteins, the density map needs to be further analyzed. The thresholding (or generally a segmentation) step is necessary to identify high density regions i.e. groups of pixels that are close in value and in space. SMLM based clustering methods often use a single, user defined, density threshold, where all values above the threshold are considered as high density regions (or clusters). Our method does not require manual selection of this important parameter. Our SOFI-based clustering analysis systematically scans over the whole range of density levels (in the density map) and quantifies the clustering behavior for each density level. This analysis provides an overview of HDR parameters (number of HDRs, average area of the HDR etc.) in relation to the density threshold, unraveling the overall clustering behavior of the sample under study. Moreover, using this information the algorithm can automatically determine an optimal density threshold to distinguish between clustering that corresponds to a random distribution of proteins and non-random behavior of proteins.

2. The Gaussian fits for the data in supplementary figure 2 do not look very convincing, in particular the double Gaussian fit in panel C. Why not try something like a multiple Poissonian distribution? Also, why are the first 4-5 data points in panel C at the edge of the std-interval?? Should they not be in the middle of the std-interval?

Thank you for noticing this issue with few data points which was caused during the export to vector graphics. We corrected the Supplementary Figure 2. We followed the advice and performed the same procedure also with the multiple Poissonian distribution. For some samples, this model provides slightly more accurate fit in terms of mean square error, but there is practically no difference on the selection of the optimal threshold and the results of the clustering analysis. For the simplicity, we decided to keep the bivariate Gaussian distribution model which has a simple physical explanation backed by our simulations. For the future applications, maybe more complex models could be tested.

3. As far as I understand, the whole analysis and in particular the simulations are based on the assumption that all molecules follow a simple on-off Bernoulli statistics in their blinking with fixed values of on- and off-switching rates. In reality, once could expected that different fluorescent molecules follow different blinking statistics, or show even multi-step blinking? How would that alter the analysis and its results?

The blinking properties of molecules can be affected by the environment and can vary spatially. We assume that the excitation within a PSF region is almost constant and the blinking rate as well as the on-state brightness vary slowly such that we can perform the calculations. The two-state model is indeed a simplification neglecting the fast blinking due to triplet state which is anyway too fast to be resolved by the camera. Different fluorophores can be in theory in slightly different microenvironments which we cannot model since we do not image on a single molecule level. Considering the resolution and sensitivity of the method, we did not observe any artifacts which could be caused by this simplification.

From the bSOFI analysis, one gets the on- and off-rates as a spatially dependent function. It would be interesting to show the images for $\rho_{on}(\mathbf{r})$ and $\rho_{off}(\mathbf{r})$.

We have added images of on-time ratio to the Supplementary Figure 6 which show the effect of illumination intensity on the photoswitching rate.

4. The presented work of using cumulants for obtaining number and brightness images has some relation to the work number-and-brightness analysis using image correlation spectroscopy (ICS) introduced by Wiseman, Gratton and Digman. It would be fair to discuss the similarities and differences that the number-and-brightness analysis by ICS has with the bSOFI approach.

We thank the reviewer for pointing to this topic. We have added this discussion into the Introduction.

Reviewer 3:

This is a brief article where the authors employed super-resolution optical fluctuation imaging (SOFI) method to analyze the clustering behaviors of membrane proteins. They examined the expression of wild type and three mutants of CD4 molecules expressed on resting T cell membranes to demonstrate the versatility of the described technology.

We appreciate well the positive evaluation of our work.

Here is a list of things recommended to be revised before publication:

1. It is unclear how much the reported technique is novel and whether it generates significant contributions to the field, as ref. 22, 24, and 25 cited in the manuscript already have demonstrated that SOFI can be applied for imaging and analysis of proteins and their structures on live cells. The introduction part of the manuscript should be revised to better highlight their unique contributions, compared to these prior works.

We revised the whole manuscript and emphasize the robustness of our approach and its novelty in providing automatized algorithm for membrane protein clustering analysis which has not been published before. The earlier works (original ref. 22, 24 and 25) demonstrated that molecular density estimation using bSOFI is feasible. Here we substantially extend this work by presenting a full novel SOFI-based clustering analysis for quantitative assessment of heterogenous protein distributions. We also show its robustness and effectiveness in comparison with the state-of-the-art SMLM-based clustering analysis (see the comment below).

2. Indeed single molecule localization microscopy (SMLM) has its own limitations in cluster analysis, which is well demonstrated in Supplementary Figure 5. However, there are some previous articles where more rigorous and prudent approaches had been developed to guarantee the integrity of cluster analysis using SMLM data, for example, Rossey, J. et al. Conformational states of the kinase Lck regulate clustering in early T cell signaling Nature Immunology 14, 82–89 (2013). I suggest the manuscript should be revised to compare the pros and cons of SMLM and SOFI based methods in more detail.

The reviewer is right that SMLM-based cluster analysis tools are evolving rapidly and recent efforts deal with the known issues (e.g. Rossey et al. *Nat Immunol* 2013, Baumgart et al. *Nat Meth* 2016; Spahn et al. *Nat Meth* 2016). Here, we are offering a novel tool based on SOFI which enables analysis of membrane protein densities (clusters) independent of an experimental setup. In Supplementary Figures 5, 6 and 7 we demonstrate robust performance of our bSOFI algorithm and compare it to the state-of-art SMLM-based imaging and clustering analysis. We show that SMLM imaging exhibits serious localization errors in dense structures (Supplementary Fig. 5). Moreover, cluster artifacts were observed in samples acquired using suboptimal illumination conditions and analyzed using SMLM-based clustering analysis (Supplementary Figs. 6 and 7). On the contrary, the data from our bSOFI approach exhibited unprecedented consistency. At optimal irradiation intensities, SMLM- and SOFI-based results were comparable. We would like to note that it is a common issue that analysis is performed on imperfect samples or in samples with varying quality of the images. We provide evidence that our method tolerates large variation in a quality

of tested samples and provides consistent results. We used previously published SMLM data of Prof. M. Sauer and colleagues (Burget et al. *Histochem Cell Biol* 2015) to avoid bias results.

Two figures were added to the Supplementary Information to highlight robustness and efficiency of our novel method:

Supplementary Figure 6: Robustness of SOFI-based clustering analysis under diverse experimental conditions.

The performance of bSOFI algorithm was compared to the state-of-the-art SMLM-based clustering analysis (SR-Tesseler [3]) using samples acquired at different irradiation intensities (0.5, 1, and 7 kW/cm²). For an unbiased comparison, this test was performed by evaluation of previously published data achieved by metabolic labelling of glycans on the basal membrane of U2OS cells by click chemistry and a Cy5 fluorophore-alkyne (see reference [2] for more details on sample preparation and image acquisition). High irradiation density (7 kW/cm²) represents favorable conditions for SMLM. Under these conditions, the results of SMLM- and SOFI-based clustering analyses are in a good agreement. Low irradiation intensity leads to inappropriate photoswitching rates. Consequently, the input image sequence exhibits high molecular density per frame and the resulting super-resolution images show artificial membrane clusters. Even with blinking correction and the state-of-the-art SMLM-based clustering analysis [3], the data demonstrate overestimation of clusters (areas, numbers) in samples with randomly distributed molecules [2]. The box plots with SMLM-based results (a) show increasing cluster area with decreasing irradiation intensity, while no such dependence was observed when SOFI-based clustering analysis was applied (b). Cluster size and total area covered by clusters (d) remain comparable for samples irradiated by various intensities and analyzed by our novel bSOFI algorithm. Correctly, SOFI density maps (c) of all tested samples show largely homogeneous distribution of molecules. On the contrary, SMLM maps (e) contain large clusters in samples with suboptimal irradiation intensity (0.5 and 1 kW/cm²). On-time ratio maps (f) reveal that lower irradiation intensity leads to higher photoswitching rates (see Supplementary Note for more details about the on-time ratio). The grayscale images in the part (c) in the lower panel represent an average intensity (left panels) and a single representative frame (right panels) from the input sequence used for image and clustering analyses (ROIs). Colorbar represents relative density ($\# \delta_{avg}$; see Supplementary Figure 1). Scale bars: 2 μ m. Of note, similar results were acquired by evaluation of the data from U2OS cells

labelled with Alexa Fluor 647-conjugated wheat germ agglutinin which binds to glycosylated structures on the cell surface (images not shown).

SMLM clustering analysis

SOFI clustering analysis

Supplementary Figure 7: Independence of SOFI-based clustering analysis on the number of input frames. The same data as in the **Supplementary Figure 6** were analyzed using 4000, 6000, 8000 and 10000 input frames (first n-frames of the sequence) for SMLM- and SOFI-based clustering analysis. SOFI results (lower panel) exhibit little or no variation with varying number of input frames. Increasing number of input frames does not influence the biased results of SMLM-based clustering analysis of data acquired under suboptimal irradiation conditions (0.5 and 1 kW/cm²; upper panel).

Importantly, our bSOFI method does not require new instrumentation. All SMLM microscopes are suitable for SOFI. Users can therefore use the same microscope and samples to acquire quantitative protein density data by our bSOFI algorithm but use advantages of SMLM to acquire better resolution under optimal experimental setup. Complementarity of SMLM and SOFI was studied in detail in the previous work (Dechout et al. *Nat Comm* 2016).

For example, can SOFI based cluster analysis (proposed method in this manuscript) deduce the absolute number of molecules within each cluster?

SOFI can deduce the absolute number of molecules, but the normalization allows us to hugely suppress influence of labelling efficiency and expression in between several measurements. Since our aim was to develop a robust clustering analysis, we adapted bSOFI with normalized values to evaluate behavior of CD4 variants at the plasma membrane of resting T cells which exhibited some variation of samples (see our comments in section 2.2 of the manuscript).

3. Did the authors verify that the expression levels of wild type and other mutant variants were equivalent to each other? In order to compare the clustering behaviors of these variants (Fig 3, Supplementary Figure 6), the expression level of these variants should be comparable to each other. One of the rudimentary way to show the comparable expression level would be showing a collection

of TIRF images acquired for the same acquisition period (up to a few seconds) for each group (or combined raw images without applying density threshold).

The reviewer is right. It was shown previously that varying labelling density influences SMLM-based clustering analysis (Baumgart et al. *Nat Meth* 2016). As requested, we calculated sum intensity for each cell ROI, plotted these values in a box plot and highlighted mean values. This figure (Supplementary Figure 8) indicates similar expression between different cell types.

Supplementary Figure 8: Comparable expression level of CD4 variants in transfected Jurkat T cells evaluated by our novel method for protein density analysis (see also **Supplementary Figure 9**). Expression levels were determined by summing up the intensity across the raw sequence of TIRF images. Median intensity was calculated in the ROI of tested samples (the same ROI as we used for the SOFI-based clustering analysis). The results reveal comparable expression level of tested CD4 variants.

4. According to the data presented in Fig 3, the palmitoylation mutant (CS1) molecules are less clustered than the truncated mutant molecules that lack the whole cytoplasmic domain (dCT). More discussion is warranted.

We are very thankful for this comment and apologize for omitting this issue in the original version of the manuscript. Indeed, dCT mutant of CD4 contains single cysteine residue (C219) which is palmitoylated. This fact was clarified in the main text (together with additional discussion of its intermediate phenotype):

'This observation is also supported by the truncation variant which lacks the cytoplasmic part (dCT) but contains single cysteine residue (C219) in the C terminus of transmembrane domain which is responsible for its palmitoylation. Reversibility of this protein lipid modification can lead to a more rapid depalmitoylation of dCT variant compared to native CD4 with two membrane-proximal cysteine residues (Supplementary Fig. 4). Indeed, dCT variant forms fewer HDRs than native CD4, but is more heterogeneously distributed compared to a non-palmitoylatable mutant (CS1).'

The presence of single palmitoylatable cysteine is also mentioned in the legend of the Supplementary Fig. 4 which schematically introduces protein variants in this study:

'dCT mutant has single cysteine residue (C219) in the cytosolic end of transmembrane domain which is responsible for its palmitoylation.'

Reviewers' Comments:

Reviewer #2 (Remarks to the Author):

The authors have significantly expanded and revised their manuscript, adding important additional and insightful information. They have answered all my questions perfectly well, and the new information makes their case for using SOFI for cluster analysis even stronger. I now strongly recommend publication as is.

Reviewer #3 (Remarks to the Author):

In this revised version, the authors indeed diligently and prudently address the reviewer's questions. First, they revised the introduction part to make an attempt to clarify what is the novel contribution of this work compared to the previous arts. Second, they performed some new analyses to directly compare the results obtained from SOFI and SMLM approaches for an identical data set. Third, some important details about the CD4 mutant experiments (expression levels, detailed sequences of each mutant) have been updated. Due to these revising efforts, the manuscript has been clearly improved.

The current manuscript well demonstrates that SOFI could be more versatile than even the state of the art SMLM technique in terms of characterizing the clustering behaviors of membrane proteins.

From the novelty point of view, the algorithm to automatically determine the optimal density threshold without relying on examiner's potential bias sounds intriguing. But the discussion/explanation on this seemingly important aspect is only one sentence (Page 3, line 204-209) with one Supplementary Fig (Suppl Fig 2). It is highly recommended that the authors further revise the manuscript to make this point more elaborated in the main text so that the novel contribution of this work in regards to this particular aspect should be more clearly communicated.

Contingent upon a satisfactory revision of the above-mentioned suggestion, the publication of this article in Nature Communications is recommended.

Response to the reviewers' comments

Reviewer 2:

The authors have significantly expanded and revised their manuscript, adding important additional and insightful information. They have answered all my questions perfectly well, and the new information makes their case for using SOFI for cluster analysis even stronger. I now strongly recommend publication as is.

We greatly appreciate this positive evaluation of our work.

Reviewer 3:

In this revised version, the authors indeed diligently and prudently address the reviewer's questions. First, they revised the introduction part to make an attempt to clarify what is the novel contribution of this work compared to the previous arts. Second, they performed some new analyses to directly compare the results obtained from SOFI and SMLM approaches for an identical data set. Third, some important details about the CD4 mutant experiments (expression levels, detailed sequences of each mutant) have been updated. Due to these revising efforts, the manuscript has been clearly improved. The current manuscript well demonstrates that SOFI could be more versatile than even the state of the art SMLM technique in terms of characterizing the clustering behaviors of membrane proteins.

We greatly appreciate this positive evaluation of the revised manuscript.

From the novelty point of view, the algorithm to automatically determine the optimal density threshold without relying on examiner's potential bias sounds intriguing. But the discussion/explanation on this seemingly important aspect is only one sentence (Page 3, line 204-209) with one Supplementary Fig (Suppl Fig 2). It is highly recommended that the authors further revise the manuscript to make this point more elaborated in the main text so that the novel contribution of this work in regards to this particular aspect should be more clearly communicated.

We appreciate reviewer's help with our manuscript. A sentence (labelled in yellow) was added to the Discussion to highlight this important aspect of our work. On the other hand, we kept relevant section in Results in its original form (lines 196-208). We prefer to balance the two options of data presentation, full-range (Fig. 3a-c) and thresholded (Fig. 3d), in this section. Both forms can provide useful information for data interpretation as can be seen for CD4 at the surface of T cells in this manuscript.

We believe that the current version of the manuscript sufficiently covers all achievements of our work.